# Ensemble Method of Convolutional Neural Networks with Directed Acyclic Graph Using Dermoscopic Images: Melanoma Detection Application

**DOI:** 10.3390/s21123999

**Published:** 2021-06-10

**Authors:** Arthur Cartel Foahom Gouabou, Jean-Luc Damoiseaux, Jilliana Monnier, Rabah Iguernaissi, Abdellatif Moudafi, Djamal Merad

**Affiliations:** 1Laboratoire d’Informatique et Systèmes, Aix-Marseille University, 163 Avenue de Luminy, CEDEX 09, 13288 Marseille, France; jean-luc.damoiseaux@lis-lab.fr (J.-L.D.); jilliana.monnier@lis-lab.fr (J.M.); rabah.iguernaissi@lis-lab.fr (R.I.); abdellatif.moudafi@univ-amu.fr (A.M.); djamal.merad@lis-lab.fr (D.M.); 2Centre de Recherche en Cancerologie de Marseille (CRCM), 27 Boulevard Lei Roure, 13009 Marseille, France

**Keywords:** ensemble method, directed acyclic graph, melanoma detection, deep learning, dermoscopic images, multiclass classification, fusion-based model, computer-aided system, skin cancer

## Abstract

The early detection of melanoma is the most efficient way to reduce its mortality rate. Dermatologists achieve this task with the help of dermoscopy, a non-invasive tool allowing the visualization of patterns of skin lesions. Computer-aided diagnosis (CAD) systems developed on dermoscopic images are needed to assist dermatologists. These systems rely mainly on multiclass classification approaches. However, the multiclass classification of skin lesions by an automated system remains a challenging task. Decomposing a multiclass problem into a binary problem can reduce the complexity of the initial problem and increase the overall performance. This paper proposes a CAD system to classify dermoscopic images into three diagnosis classes: melanoma, nevi, and seborrheic keratosis. We introduce a novel ensemble scheme of convolutional neural networks (CNNs), inspired by decomposition and ensemble methods, to improve the performance of the CAD system. Unlike conventional ensemble methods, we use a directed acyclic graph to aggregate binary CNNs for the melanoma detection task. On the ISIC 2018 public dataset, our method achieves the best balanced accuracy (76.6%) among multiclass CNNs, an ensemble of multiclass CNNs with classical aggregation methods, and other related works. Our results reveal that the directed acyclic graph is a meaningful approach to develop a reliable and robust automated diagnosis system for the multiclass classification of dermoscopic images.

## 1. Introduction

Skin cancers are the most common types of cancer in the Caucasian population [1]. Melanoma is the most lethal skin cancer due to its possible evolution into metastasis [1]. Among pigmented lesions, it is particularly difficult to differentiate in melanoma between nevi and seborrheic keratosis [2,3,4]. Typical pigmented melanoma, nevi, and seborrheic keratosis can be distinguished easily. Figure 1 depicts these lesions and a typical melanoma (Figure 1a), a typical nevi, (Figure 1b), and typical seborrheic keratosis (Figure 1c), which do not raise any diagnosis issues for dermatologists. However, atypical nevi or seborrheic keratosis can be confused with melanoma. Figure 1d–f show some atypical melanoma, nevi, and seborrheic keratosis and highlight how it can be challenging for a dermatologist to rule out a melanoma among these types of pigmented lesions. Faced with atypical pigmented lesions, dermatologists require excision with histological analysis to confirm or reject a diagnosis of melanoma.

These types of atypical pigmented lesions explain the high ratio of the number of lesions excised to the number of melanomas diagnosed [6]. A total of 9.6 suspicious benign lesions are excised before reaching a confirmed diagnosis of melanoma [7]. Each excision can lead to scarring and post-surgery complications. The principal objective for dermatologists is to decrease this number and excise only true melanomas. Thus, differentiating early melanoma from nevi and seborrheic keratosis not only constitutes a daily problem for dermatologists, but also has the potential to decrease cancer deaths since melanoma can be cured with a simple excision at an early stage [8].

Most dermatologists are currently using a dermoscopic sensor during dermatological examination for skin cancer screening. It is a non-invasive dermatological tool allowing the visualization of the lesions’ patterns and structures with a high resolution. It involves a magnification lens and different lighting schemes, such as non-polarized and polarized light. Polarized light helps to minimize the light reflection of the skin’s surface and highlights the detailed patterns and vascularization of the lesion. A dermoscopic sensor helps dermatologists to recognize specific features for the early diagnosis of skin cancer that are sometimes not visible to the naked eye [9]. Figure 2 illustrates the use of a dermoscopic sensor.

The contribution of dermatoscopy has been investigated by many authors and dermatologists [10,11], demonstrating its efficiency in increasing melanoma diagnostic accuracy by 5% to 30% over clinical visual inspection alone. Frequent skin cancer screening of the general population with a dermoscopic examination of pigmented lesions is necessary to detect early melanoma; unfortunately, the lack of dermatologists prevents the development of large screening programs. Therefore, CAD systems have been developed to assist dermatologists to achieve the early diagnosis of melanoma.

To help dermatologists diagnose melanoma early and to reduce the number of unnecessary excisions of benign lesions, the computer vision community has developed several CAD systems. A CAD system is an automatic tool used to support dermatologists in their diagnosis. Before 2015, CAD development was mainly based on handcrafted features. It consisted of extracting features such as shape, color, and texture. These approaches were inspired by the ABCDE criteria (A, asymmetry; B, irregular borders; C, inhomogeneous color; D, diameter > 6 mm; and E, evolution) [12]. The extracted characteristics were then used as input vectors for a machine learning algorithm (multilayer perceptron (MLP), SVM, KNN, logistic regression, etc.). Celebi et al. proposed an approach to classify dermoscopic images involving border detection and handcrafted extraction of features (texture, color, and shape). These features were then used to train an SVM for classification with feature selection [13]. However, the ABCDE criteria are not the best features to use for melanoma detection [14]. Moreover, these features are assimilated to low-level features in CAD systems, which can limit the overall accuracy of the CADs. CNNs attempt to model high-level abstractions in data using multiple processing layers. Due to the availability of public datasets and the advances in computing capacity, there is a growing trend in their use in skin lesion classification. Esteva et al. [15] were the first to compare CNNs’ diagnostic accuracy with that of dermatologists. They found that most dermatologists, and especially the less experienced ones, were outperformed by CNNs.

The computer vision community relies on the ensemble method to achieve highly accurate performance in the multiclass classification of skin lesions. The ensemble method is based on fusing a finite set of classifiers [16]. Harangi et al., for example, combined the output of the classification layer from four CNNs using a weighted majority voting strategy for a three-class classification task [17]. Pacheco et al. tested different approaches including simple majority voting, maximum probability, and the average of probabilities to merge the output of 13 CNNs in an eight-class classification task [18]. The average probability achieved the best results. Mahbod et al. proposed a framework based on three CNN backbones, where each model was trained on images of skin lesions of six different sizes, ranging from 224 × 224 to 450 × 450 pixels. All the models constructed were then assembled on a three-level ensemble strategy based on the average of predicted probabilities [19]. In [20,21,22,23,24], the average of probabilities has also been used as an aggregated method to improve the performance of CAD.

Broadly, current studies applying ensemble methods follow a similar workflow. First, several multiclass CNNs are trained for a specific task and then their outputs are merged using an aggregation approach. An overview of related works applying ensemble methods is provided in Table 1.

The most used aggregation methods are:Max-Win strategy: The class selected by the Max-Win strategy is the class that receives the maximum number of votes.Product of probabilities strategy: The product of the individual outputs of the CNNs is calculated and the selected class is determined by the maximum of the normalized products.Average probability strategy: The arithmetic mean of the confidence values of each CNN is calculated, and the selected class is determined by the maximum of the normalized means.Max confidence strategy: The class selected by the max confidence strategy is the class that received the maximum confidence score.Geometric mean strategy: The geometric mean of the confidence values of each CNN is calculated, and the selected class is determined by the maximum of normalized means.

The multiclass classification of pigmented lesions remains a challenging task because skin lesions have a high degree of similarity, making their classification a complex task that requires an extensive amount of labeled data and careful definition of the network’s free parameters to train an accurate CNN. Additionally, CNNs behave as black boxes, making it difficult for dermatologists to interpret their prediction.

Rather than simply merging several multiclass CNNs, as has often been the case in most work using the ensemble method, an innovative approach involves decomposing the initial multiclass problem into several less complex classification tasks. Galar et al. stated that multiclass classification is typically more difficult than binary classification [26]. They explained that the decision boundary of a multiclass classification problem tends to be more complex than a binary classification problem. Therefore, researchers have investigated decomposition and ensemble methods as an alternative to resolve these problems. The idea behind the decomposition and ensemble method is to split the multiclass problem into a set of binary problems and then aggregate the results. The two well-known approaches to developing a decomposition and ensemble strategy are one-versus-rest and one-versus-one [26]. For an N-class classification, each approach is described as follows:The one-versus-rest approach consists of constructing a set of N binary classifiers. Each classifier is trained with one class as the positive and all the others as the negatives. The final decision corresponds to the class associated with the classifier with the highest output value.The one-versus-one approach consists of constructing all possible binary classifiers from the N classes. Each classifier is trained on only two classes out of the N initial classes. Thus, there will be N(N – 1)/2 classifiers. The outputs of these base classifiers are combined to predict the final decision.

The main limitations of these approaches are that the one-versus-one approach tends to overfit the overall N-class classifier, and the Max-Win algorithm used does not have bounds on the generalization error [27]. Therefore, to remedy these disadvantages, Platt [27] proposed a decision-tree-based pairwise classification called the decision directed acyclic graph (DDAG). Platt demonstrated that DDAGs provide good generalization performance and their structure is efficient to train and evaluate.

In this study, we relied on the decomposition and ensemble method to develop an accurate automated diagnosis of melanoma, nevi, and seborrheic keratosis. For this purpose, we constructed a novel ensemble of CNNs based on DDAGs. We hypothesized that decomposing a multiclass problem into a binary problem would reduce the complexity of the initial multiclass problem faced by CNNs and simultaneously increase the overall performance. The DDAG follows a hierarchical workflow mimicking the multi-step reasoning used by dermatologists faced with pigmented lesions to make a diagnosis [28]. Thus, following a hierarchical structure can ensure that CAD decision-making is understandable for dermatologists and increase their use in a clinical setting. To the best of our knowledge, this is the first attempt to use a DDAG as a decomposition and ensemble strategy with CNNs. The main contributions of this work are:Decomposing the initial multiclass classification of pigmented lesions into a binary problem to reduce the complexity of the task and increase the overall classification performance;Using a directed acyclic graph as an ensemble method to perform multiclass classification with CNNs;Following a hierarchical workflow provides more transparent decision-making of the computer-aided diagnosis system, thus making it more understandable for dermatologists.

The remainder of this paper is organized as follows: Section 2 describes the methods applied. In Section 3, we present the results of the experiments conducted on the 2018 International Skin Imaging Collaboration (ISIC) public dataset, and we discuss in detail the results of our proposed method. Finally, we conclude the work and discuss its future scope in Section 4.

## 2. Materials and Methods

In this section, we provide a detailed description of our proposed approach to build a computer-aided diagnosis system to differentiate melanoma, nevi, and seborrheic keratosis. We selected these three most challenging classes in melanoma detection based on our collaborative work with onco-dermatologists specialized in melanoma management (screening and treatments). The section is divided into three subsections. The first subsection is devoted to describing the dataset, its preparation, and the preprocessing applied to images. The second subsection presents the theory behind the DDAG and describes the architecture and the methodology used to train our models. Finally, the third subsection defines the metrics used to evaluate our model. The flowchart of the proposed framework is illustrated in Figure 3.

### 2.1. Dataset

We evaluated our approach on the common problems faced by onco-dermatologists [2,3,4]: the early diagnosis of melanoma amongst nevi and seborrheic keratosis. We performed this task using the ISIC 2018 public dataset [5].

#### 2.1.1. Dataset Preparation and Class Balancing

The selected dataset comprises 1113 melanomas, 6705 nevi, and 1099 seborrheic keratosis. We randomly split the original dataset into 80% as the training set and 20% as the test set. Then, we proceeded to cross-validation to evaluate our model; we randomly split the previously established training set into three groups each containing a training set (80% of the original training set) and a validation set (20% of the original training set). To alleviate the imbalanced distribution of classes in our training set, we used artificial data generation of images on the training set for each split. Perez et al. [29] demonstrated the positive impact of using data generation for training melanoma classification models. The methods selected to generate the artificial data were horizontal flipping, vertical flipping, rotation, width, and height shift. Table 2 describes the distribution of the dataset and the proportion of generated images used for fine tuning.

#### 2.1.2. Image Preprocessing

In our study, we kept the preprocessing to a minimum to facilitate the reproducibility of our CAD. We applied standard preprocessing for deep learning classification, namely normalization, cropping, and image resizing. We also performed color standardization to ensure the robustness of our algorithms. The images in the dataset [5] were collected from multiple sources and acquired under different setups. This is illustrated in Figure 4, which shows the variation in the illumination from one image to another.

Thus, we used the gray world [30] algorithm to perform color standardization of the images. As advised [31], we modified the original algorithm by pre-segmenting the image and computing the average color of each patch. We applied Equation (Equation 1), where *I* represents a color image, $e_{c}$ is the illuminant of each component, c∈{R,G,B}, and *n* is the number of patches in the image (for details, refer to [31]). This is particularly important, as [32] reported a substantial benefit of this type of preprocessing in skin lesion classification.
(1)$$\frac{\int \sum_{j=1}^{n}I_{c}\left(x_{j}\right)dx_{j}}{n}=ke_{c}$$

Images were normalized by subtracting the mean RGB value of the ImageNet dataset, as suggested in [33]. This was performed to facilitate the training of the models.

We also resized images to fit them to the required input size (224 × 224 pixels) of the pretrained CNNs used in our implementation. For this, we first center-cropped the images to 450×450 pixels (from their original size of 600×450 pixels) to preserve the aspect ratio. The size of 450×450 pixels allowed us to obtain the entire lesion present in the images. Additionally, the cropping patch was programmed so that its center coincided with the center of the image, to ensure that we were recovering the entire lesion. This choice is justified by almost all the lesions in the dataset being located in the center of the image. Then, we resized our images to 224 × 224 pixels using a bicubic interpolation Figure 4 depicts the appearance of the images after being preprocessed.

### 2.2. Methodology

#### 2.2.1. Convolutional Neural Network

Several CNNs have been reported. Some of them are available as pretrained models, trained on 14 million images from the ImageNet dataset. Thus, we can reuse their weights and biases, and fine tune these models in order to apply them to specific classification tasks; this is known as transfer learning. We tested our framework with three well-known pretrained CNNs that have been successfully used in the task of classifying skin lesions: VGG network [33] and residual neural network (ResNet) [34].

ResNet architecture: A deeper network leads to saturation of accuracy because the gradient of the loss function rapidly approaches zero during backpropagation, making it difficult for the network to learn; this is called the vanishing gradient issue. The main idea of ResNet is to reduce the vanishing gradient with the help of a residual block (see Figure 5). The original implementation of ResNet [34] has several variations. In our work, we used ResNet50, which has 50 convolutional layers with filters of 7 × 7, 3 × 3, and 1 × 1. Convolutional layers are grouped as residual blocks to construct the entire architecture. Each residual block consists of a few stacked layers of convolutional layers, a zero padding layer, a batch normalization layer, rectified linear unit layers as an activation function, and a max pooling layer. A global average pooling layer ends the residual blocks to condense the output feature maps into a feature vector, followed by fully connected layers as a classifier. We modified the ResNet50 architecture by replacing the output layer with a new, fully connected (FC) layer of 2 nodes to perform binary classification. The modified ResNet50 is shown in Figure 6 (top). For simplicity, we refer to this architecture as ResNet_2.

The VGGNet architecture: VGGNet is a well-documented and commonly used CNN architecture in computer vision. Several variations of VGGNet were initially proposed [33]. The variants differ in terms of the depth of the network, ranging from 16 to 19 layers. We selected the VGG16 and VGG19 architectures, which have previously shown high-quality performance on the skin lesion classification task. Both VGG16 and VGG19 expect a 224 × 224 size image as input. They consist of five convolutional blocks. Each convolutional block consists of two, three, or four convolutional layers with a filter of size 3 × 3, rectified linear unit layers as activation function, and a max pooling layer. The networks are concluded with a classifier block consisting of three FC layers. In our work, we modified the original VGG16 and VGG19 by removing their last FC layer and replaced each of them with a FC layer with 2 nodes. The modified VGG16 and VGG19 are presented in the middle and bottom of Figure 6, respectively. In the following section, these architectures are denoted as VGG16_2 and VGG19_2, respectively.

#### 2.2.2. DDAG Theory

A DDAG is a graph whose edges have an orientation and no cycles. The DDAG algorithm was initially introduced by Platt [27] to extend SVM to perform multiclass classification. The idea of DDAG combines a set of binary classifiers into a multiclass classifier. The main advantage of DDAG for the ensemble method is reducing the training and evaluation time, using fewer computer resources while maintaining accuracy, compared with the classical aggregate method.

For adequate formalization, we considered a binary CNN to be a function $S_{ij}:x\Rightarrow R^{2}$, which assigns two confidence values $p_{i},p_{j}\in R$ to a new, formerly unseen image *x*, where $p_{i},p_{j}\in \left[0,1\right]$ and $\sum p_{i}+p_{j}=1$. Each $S_{ij}$ classifies images according to whether they belong to class $c_{i}$ or $c_{j}$. i,j indicates the nature of the lesion, in our case i,j=B;M;N, where *B* indicates benign keratosis, *M* indicates mlelanoma, and *N* is nevi. Figure 7 shows the DDAG for the 3-class classification problem. In Figure 7 $\bar{i}$ denotes that *x* does not belong to class *i*. Suppose that there are K classes, the DDAG contains K(K − 1)/2 binary classifier. For a K-class classification problem, K − 1 nodes are evaluated to derive a decision. The path taken to reach the final decision on the DDAG is known as the evaluation path. Algorithm 1 describes the steps followed to determine the DDAG structure for the 3-class problem while classifying observation *x*.
**Algorithm 1** DDAG structure.**Require:** Image *x*, 3 pairwise CNNs $S_{ij}$, list of the three classes class_list=[1,2,3] **while** len(class_list) >1 **do**  Select two elements *i* and *j* in class_list  Generate the prediction of the class associated to *x* with $S_{ij}$  **if**
$S_{ij}$ associates *x* to class $c_{i}$ **then**   Remove *j* from class_list  **else**   Remove *i* from class_list  **end if** **end while** Predict that *x* belongs to the class represented by the only element remaining in class_list

#### 2.2.3. Aggregation Functions Theory

Aggregation functions are mathematical tools with the ability to combine multiple attributes into one single output. More precisely, an *n*-dimensional aggregation function is a monotonic function $f:{\left[0,1\right]}^{n}\Rightarrow \left[0,1\right]$ that satisfies the boundary condition f(0,⋯,0)=0 and f(1,⋯,1)=1. In the ensemble method of convolutional neural networks, the classical aggregation functions used are the arithmetic mean (avg), geometric mean (gmean), product functions (prod), and maximum confidence score (mconf). To compare our approach with these classical aggregation methods, we modified the CNNs previously described in Section 3.2 (ResNet50_2, VGG16_2, and VGG19_2). For this purpose, we replaced the last layers (FC-2) of the previous binary classifierswith an FC layer with 3 nodes to perform three-class classification (ResNet50_3, VGG16_3, and VGG19_3). This comparison is analyzed in Section 3.5. Let $p_{i,j}$ denote the confidence value assigned by the *j*th three-class classifier to the *i*th class, and let $p_{i}^{{}^{\prime }}$ be the probability, derived from the confidence scores of CNNs constituting the ensemble, that an input image *x* belongs to class *i*, i∈ {melanoma, nevi, seborrheic keratosis} and j∈ {ResNet50_3, VGG16_3, VGG19_3}. The formulation of $p_{i}^{{}^{\prime }}$ depending on the aggregation method is:avg:
(2)$$p_{i}^{{}^{\prime }}=\frac{\sum_{j=1}^{m}p_{i,j}}{m}$$prod:
(3)$$p_{i}^{{}^{\prime }}=\frac{\prod_{j=1}^{m}p_{i,j}}{\sum_{i=1}^{n}\prod_{j=1}^{m}p_{i,j}}$$mconf:
(4)$$p_{i}^{{}^{\prime }}=\frac{\max_{j}p_{i,j}}{\sum_{i=1}^{n}\max_{j}p_{i,j}}$$gmean:
(5)$$p_{i}^{{}^{\prime }}=\frac{\sqrt[{m}]{\prod_{j=1}^{m}p_{i,j}}}{\sum_{i=1}^{n}\sqrt[{m}]{\prod_{j=1}^{m}p_{i,j}}}$$max-win:
(6)$$p_{i}^{{}^{\prime }}=\frac{\sum_{j=1}^{m}F\left(p_{i,j}\right)}{m}$$
(7)$$where\;F\left(p_{i,j}\right)=\left\{\begin{array}{c} 1,\;if\;p_{i,j}=\max_{j}p_{i,j}\\ 0,\;otherwise \end{array}\right.$$

#### 2.2.4. Model Training

We used the Adam [35] optimizer to update the weights and biases of our networks at every iteration to minimize the loss function output. We calculated the loss value of the models using a weighted binary cross-entropy function. The general term of the cross-entropy loss is:(8)$$L=-w_{i}\sum_{n=1}^{N}plog\left(q\right)$$
where *p* is the ground-truth label, *q* is the predicted SoftMax probability, $w_{i}$ is the weight for class *i*, and *N* is the number of classes. We weighted the loss function with the inverse normalized frequency of each class defined as follows:(9)$$w_{i}=\frac{N}{n_{i}}$$
where $n_{i}$ represents the number of samples for class *i*. Our network weight was initialized with ImageNet’s pretrained weights. For each model, we tested different hyperparameters for 150 epochs during training. More precisely, the hyperparameters that we tested were the initial learning rate, varying from 0.01 to 0.0001, and the percentage of the last layers of the network required to fine tune, varying from 4% to 65%. Additionally, we adapted the dynamic learning (scheduled_lr) by using a polynomial decay schedule of the initial learning rate (lr) if the loss error of the validation did not decrease after 8 epochs. Thus, the new learning rate is:(10)$$scheduled\_lr=lr*(1-\left(\frac{current\_epoch}{total\_epochs}\right))$$
Table 3 summarises the hyperparameters search space used to finetune our models. During finetuning, we also added a condition to stop the training earlier when the accuracy on the training set exceeded the accuracy on the validation set by more than 10%, and based on the models’ checkpoint, we selected the saved model obtaining the best balanced accuracy score. This was performed to avoid overfitting.

All our experiments were conducted using a system with a 3.2 GHz processor, 16 GB of memory, and a Nvidia GeForce Rtx 2080 GPU card. We used MATLAB 2020 to center-crop and apply color constancy to our data. The keras library with tensorflow as the backend was used to train our models. The code for our experiments is publicly available at Appendix A.

### 2.3. Performance Criteria

To allow application of our method in a clinical context, we used various metrics to evaluate our framework. This was performed by calculating the area under the receiver operating characteristic curve (AUROC) and the balanced accuracy of the classification. Although the first metric is well-known in the community, the balanced accuracy is much more recent and was introduced during the 2018 skin image analysis challenge [5]. We used balanced accuracy metrics to evaluate the CNN performance despite the prevalence of benign lesions in our dataset. The sensitivity and the balanced accuracy were calculated based on the generated confusion matrix of our models. The confusion matrix provides information on true positive (*TP*), true negative (*TN*), false negative (*FN*), and false positive (*FP*) predictions. The formulations of each of these metrics are:(11)$$Balanced\;accuracy\;\left(BACC\right)=\frac{sensitivity+specificity}{2}$$
(12)$$Sensitivity\left(S\right)=\frac{TP}{TP+FN}$$

To measure these indexes, we converted the classification probability vectors to binary classification vectors using a threshold of 0.5.

### 2.4. Statistical Analyses

We performed non-parametric statistical tests. A paired t test was employed to compare two models. In cases where more than two comparisons were carried out, we used Kruskal–Wallis’s test, and afterward, a post hoc multiple-comparison test using Dunn’s test was employed. Results were considered statistically significant if *p*-value < 0.05. Statistical calculation and visualizations were carried out using GraphPad Prism, version 5.03.

## 3. Results and Discussion

We evaluated our novel approach based on the combination of DDAG and binary CNNs. First, we tested the performance of the individual binary classifiers. Second, we analyzed the effect of varying the root node after aggregating the outputs with the DDAG approach. Third, we compared the result of our method with three well-known CNN architectures on a three-class classification task. Then, we analyzed the performance of our best DDAG structure. Finally, we evaluated our approach against other conventional aggregation strategies. We used a three-fold cross-validation on the training set and present the average and standard deviation for the BACC, the sensitivity (S), and the AUROC. In the following, we refer to melanoma, nevi, and seborrheic keratosis as MEL, NEV, and SEK, respectively.

### 3.1. Performance of Binary CNNs

Table 4 shows the results obtained with a three-fold cross-validation on the training set with resnet50_2, VGG16_2, and VGG19_2 for each individual task: MEL versus NEV, MEL versus SEK, and NEV versus SEK. The BACC and the sensitivity for each class are presented. Mostly, we observed that the classifiers performed very well in binary classification. For our task, we observed that the backbone model VGG19 performed better than Resnet50 and VGG16. These results can be explained by the deeper VGG19 architecture compared with VGG16 and Resnet50. Therefore, VGG19 can learn more discriminating features. Interestingly, among these three tasks, seborrheic keratosis and nevi were easiest to distinguish, with the best performance obtained by the binary CNNs NEV vs. SEK. Seborrheic keratosis is very dark and composed of completely different patterns to melanocytic lesions, such as keratin structures, horn cysts, or a cerebriform pattern. However, melanoma can be confused with seborrheic keratosis; melanoma can be very dark, similar to seborrheic keratosis, and can sometimes mimic seborrheic keratosis by having atypical structures. The most challenging tasks for our framework are distinguishing benign melanocytic lesions (nevi) from malignant melanocytic lesions. When the melanoma is excised at an early stage with a thin Breslow (thickness of the melanoma), the difficulty of differentiating melanoma from nevi is high even for dermatologist experts. Moreover, some melanoma are raised on nevi, so they may share the same patterns and structures (reticular pattern or dotted pattern); however, for melanomas, the pattern is more irregular than that of nevi.

### 3.2. Impact of Root Node

The second aspect that we investigated was the effect of variations in the root node on the overall performance of our approach. This was conducted based on the BACC. The results of this analysis are presented in Table 5. Regardless of the type of DDAG, we noticed that the overall performance of the framework depends on the performance of each individual classifier, explaining why DDAGs based on VGG19 performed better. The DDAG structure based on VGG19 reached BACCs between 73.7% and 76.6%, compared with the 72.55–73.25% for the VGG16 and 70.1–71.1% for ResNet50 backbone models. Moreover, the choice of the DDAG structure may slightly affect the final accuracy of the framework, which is similar to the observation of [36] with support vector machine. Thus, inspired by [37], the optimal structure of the DDAG was obtained by placing the classifier with the greatest generalization ability in the root node. This explains why DDAG structures with the SEK vs. NEV classifier as the root node performed better on VGG19_2 and VGG16_2, and the best performance was achieved with root MEL vs. NEV for resnet50_2 (Section 3.1). The best structure with the most accurate performance (BACC = 76.6 ±0.39%) was obtained with the DDAG structure based on VGG19 and having a binary CNN on the task with NEV vs. SEK as the root node.

### 3.3. Multiclass CNNs Versus DDAG Model

Table 6 shows the evaluation of our main hypothesis. We compared our approach with ResNet50, VGG16, and VGG19 trained on a three-class classification task based on the results of the 3-fold cross-validation. For a faithful comparison, only the classification layer was modified to adapt it to a three-class classification (Section 2.2.3). We refer to these adapted models as ResNet50_3, VGG16_3, and VGG19_3. The DDAG-based approach achieved the best BACCs compared to multiclass CNNs.

The best models obtained for each configuration were then selected and evaluated on the test set for an in-depth analysis. We observed that the DDAG structure with the BEK vs. NEV classifier as the root node and the VGG19 architecture as the backbone model obtained the best performance, reaching a balanced accuracy of 76.6% on the test set. However, we highlight that the performance of the DDAG structure is closely linked to the choice of the backbone model, as illustrated by our results obtained with VGG16 and ResNet50. Models with potentially better performance, such as EfficienNet [38] and SeNet [39], may improve the accuracy of the DDAG structure.

On the other hand, binary CNNs aggregated with a DDAG structure achieved better performance than 3-class CNNs. These results matched with the previous analysis (see Table 6). We performed statistical analyses using a paired *t* test on the predicted probabilities of each model and, interestingly, we found that scores from the DDAG models were significantly different from those of multiclass CNNs (Table 7). We thus concluded that decomposing a multiclass problem into a binary problem reduces the complexity of the initial problem and increases the overall performance.

### 3.4. Performance Analysis of Our Best DDAG Structure

Figure 8 shows the receiver operating characteristic curves obtained by our best DDAG structure for each lesion in the test set. Our framework achieved an AUROC of 0.93, 0.87, and 0.88 for seborrheic keratosis, melanoma, and nevi, respectively. We observed that melanoma remained the most challenging class.

We presented the structure of our framework to a dermatologist for an in-depth analysis. To facilitate the dermatologist’s analysis, we associated each prediction provided by a classifier and its corresponding heatmap, allowing visualization of the regions contributing to the prediction; heatmap generation was implemented with the Grad-CAM method [40]. Figure 9 illustrates the decision strategy of our best DDAG structure. As an example, we present a challenging pigmented lesion that was classified as a melanoma at the end of this framework. The arrows in green represent the evaluation path in this case. The dermoscopic image (input image in Figure 9) shows a pigmented lesion that is slightly suspicious. The reticular network is irregular and enlarged on the left part. On this part and in the middle, we can also observe a blue white veil color with some dots corresponding to a regression area, which is associated with melanoma diagnosis. Interestingly, the heatmap shows the decision-making area of the CNN, focusing its prediction on the atypical left part of the lesion, the most suspicious for melanoma diagnosis.

### 3.5. Comparison with Other Methods

We further compared our approach based on DDAG with commonly used aggregation methods (avg, mconf, prod, gmean, and max-win). For this, the best models obtained with ResNet50_3, VGG16_3, and VGG19_3 during cross-validation were merged following these aggregation methods. The results presented in Table 8 summarize for each method the performance obtained on the test set and highlight the outcome of Kruskal–Wallis’s test and post hoc multiple-comparison on the predicted probabilities. Here, “g.r” denotes the group rank of methods with stastistically similar predicted scores, and “s.o.g” is the set of other groups that are statistically worse. An empty set indicates that a particular method was not statistically better than any other group.

The DDAG structure achieved the best BACC (76.6%) amongst the ensemble of multiclass CNNs with classical aggregation methods. Moreover, the probability scores generated by our approach were statistically different (*p* < 0.05) from those of other classical aggregation methods, which confirms the robustness of our DDAG structure and its ability to improve the performance of a computer-aided diagnosis system. We also found that, among the classical aggregation methods, avg, max_conf, and gmean achieved the best performance, with no statistically difference amongst their predicted score. These results suggest that these are the best of the classical aggregation approaches to use for building a CAD on dermoscopic images. The product of the probabilities strategy was the worst performer. Thus, the product significantly enhances the propagation of the worst prediction probabilities. To reduce this effect in the application of ensemble methods, we recommend merging only classifiers with similar performance.

We also compared our method with existing methods on the same three-class classification task [41,42]. Based on the BACC, our approach outperformed these related methods.

Our approach is much simpler to interpret by dermatologists because it follows a hierarchical workflow similar to two-step reasoning [28], whereas conventional approaches simply aggregate several CNNs without providing transparency in the decision-making process.

## 4. Conclusions and Future Work

In our research, we implemented a new CAD framework on dermoscopic images for multiclass classification of melanoma, nevi, and seborrheic keratosis. Detecting melanoma among these two classes is a challenging daily task for dermatologists. We introduced a novel ensemble method of convolutional neural networks inspired by the decomposition and ensemble method. This approach is based on a set of three binary CNNs trained to differentiate one of the three lesions from another lesion (one-versus-one approach). Then, CNN outputs are aggregated using the DDAG. Based on our results, this approach helps the method to easily outperform a multiclass CNN. We further compared our framework with current ensemble methods: arithmetic mean, simple majority voting, maximum confidence score, geometric mean, and product of the probabilities. We demonstrated that our approach outperformed all the classical aggregation methods. These results highlighted the effectiveness of the proposed method. Our study corroborates that decomposing a multiclass problem into a binary problem reduces the complexity of the initial multiclass problem for CNNs and therefore increases the accuracy of the CAD. Notably, the proposed approach follows a hierarchical workflow, which provides transparency in the decision-making process and thus facilitates their interpretation by dermatologists. However, the overall performance of the CAD depends on the accuracy of the pairwise CNNs in the framework. Therefore, further investigations should include the performance of each individual CNN in the decision’s thresholds, which may alleviate their effect on the performance of the CAD.

## Figures and Tables

**Figure 1 sensors-21-03999-f001:**
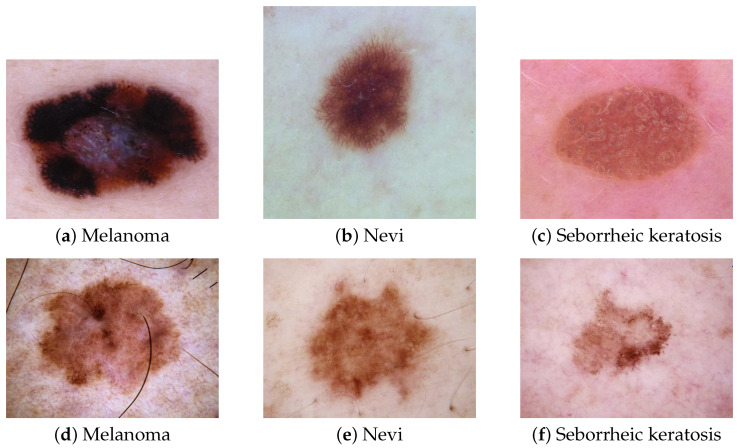
Pigmented skin lesions taken from ISIC [5] dataset. Typical lesions (**a**) melanoma, (**b**) nevi, and (**c**) seborrheic keratosis do not pose any diagnosis issues for dermatologists, whereas atypical lesions (**d**) melanoma, (**e**) nevi, and (**f**) seborrheic keratosis, having poor intra-lesion features, are much more challenging to differentiate.

**Figure 2 sensors-21-03999-f002:**
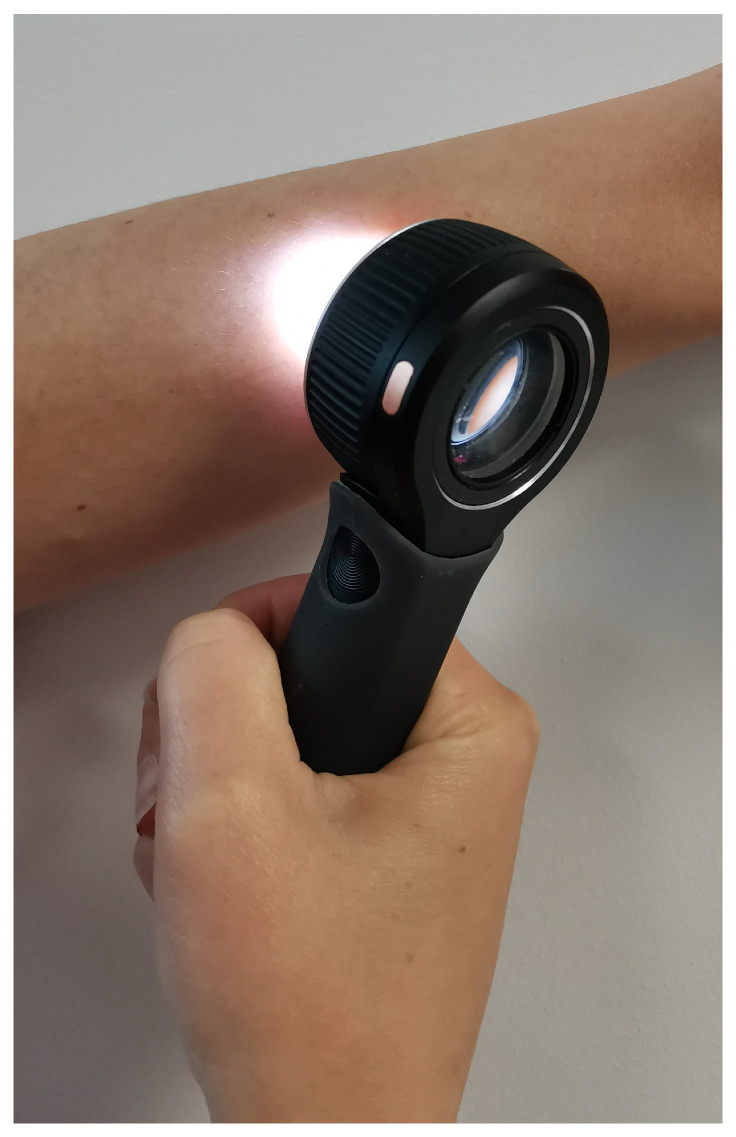
Photograph of a handheld dermoscopic sensor with non-polarized and polarized light, which is used by dermatologists during their clinical examination. The dermatoscope shown in the figure is produced by Dermoscope DermLite DL4, 3GEN Inc., San Juan Capistrano, CA, USA.

**Figure 3 sensors-21-03999-f003:**
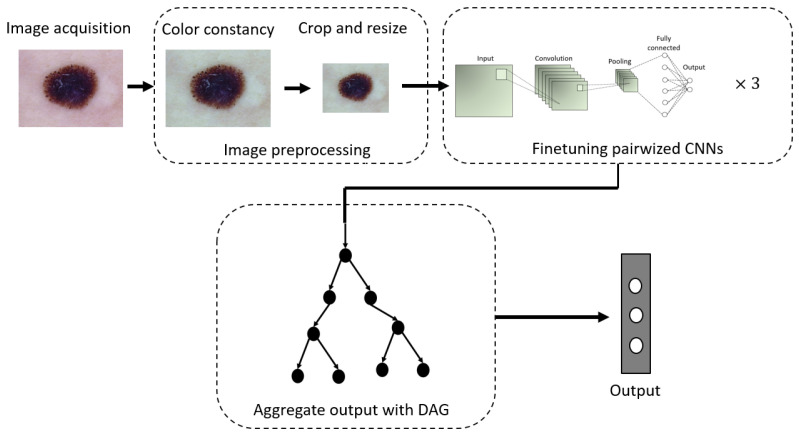
Block diagram of the proposed computer-aided diagnosis system. Skin images are first preprocessed. Then, three binary CNNs are trained using a one-versus-one approach to differentiate lesion *i* from another lesion *j*. Finally, the output of each CNN is aggregated using the directed acyclic graph (DAG) to output the final prediction.

**Figure 4 sensors-21-03999-f004:**
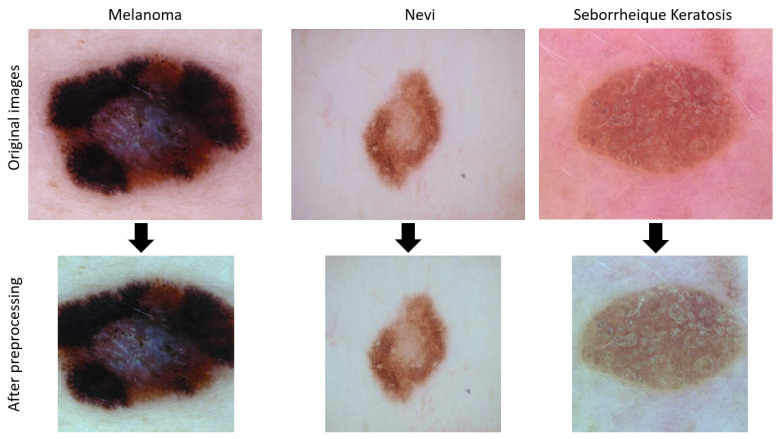
A sample of images in the dataset before and after preprocessing.

**Figure 5 sensors-21-03999-f005:**
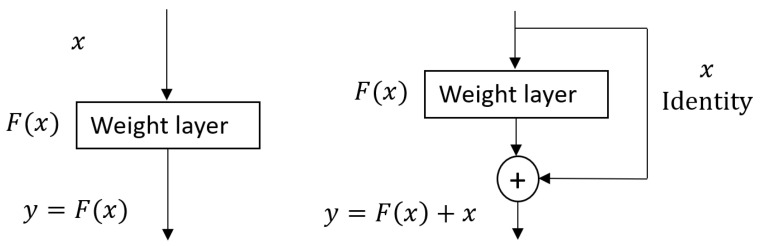
An illustration of a plain block (**left**) and a residual block (**right**).

**Figure 6 sensors-21-03999-f006:**
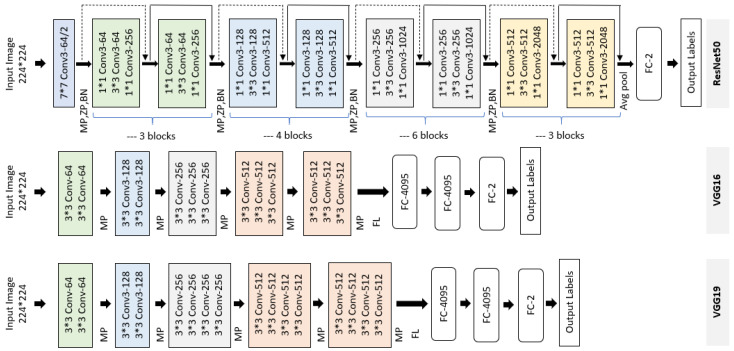
The structures of the convolutional networks used in our method. (**top**) The modified ResNet50, (**middle**) the modified VGG16, and (**bottom**) the modified VGG19. In both cases, we replaced the last fully connected (FC) layer with an FC layer with 2 nodes. MP and FL represent the max pooling and flattening layers, respectively.

**Figure 7 sensors-21-03999-f007:**
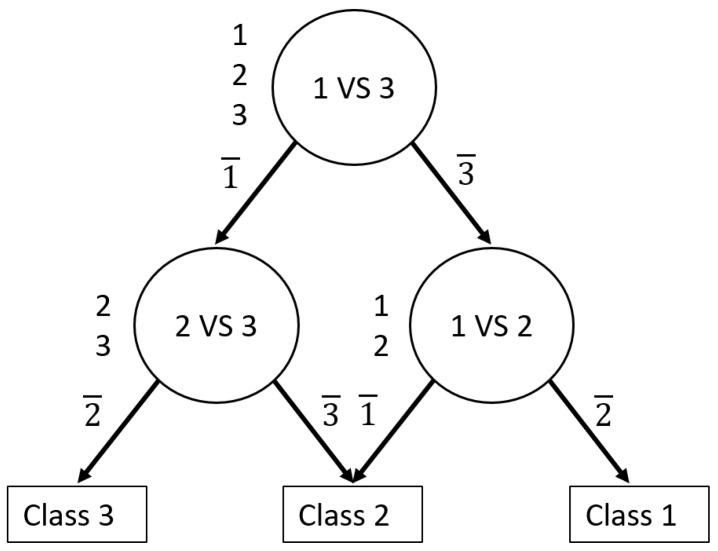
The decision directed acyclic graph (DDAG) for finding the best of three class.

**Figure 8 sensors-21-03999-f008:**
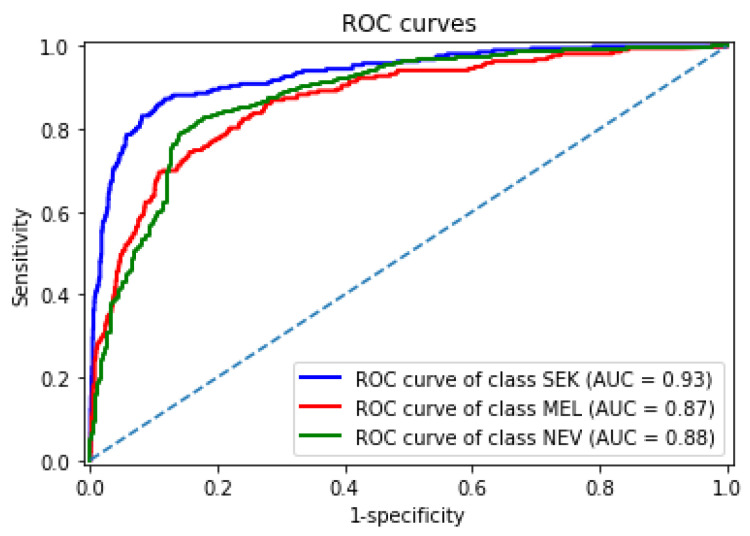
Receiver operating characteristic (ROC) curves of our best model. The area under the curve (AUC) of the ROC is provided for each lesion class: melanoma (MEL), seborrheic keratosis (SEK), and nevi (NEV).

**Figure 9 sensors-21-03999-f009:**
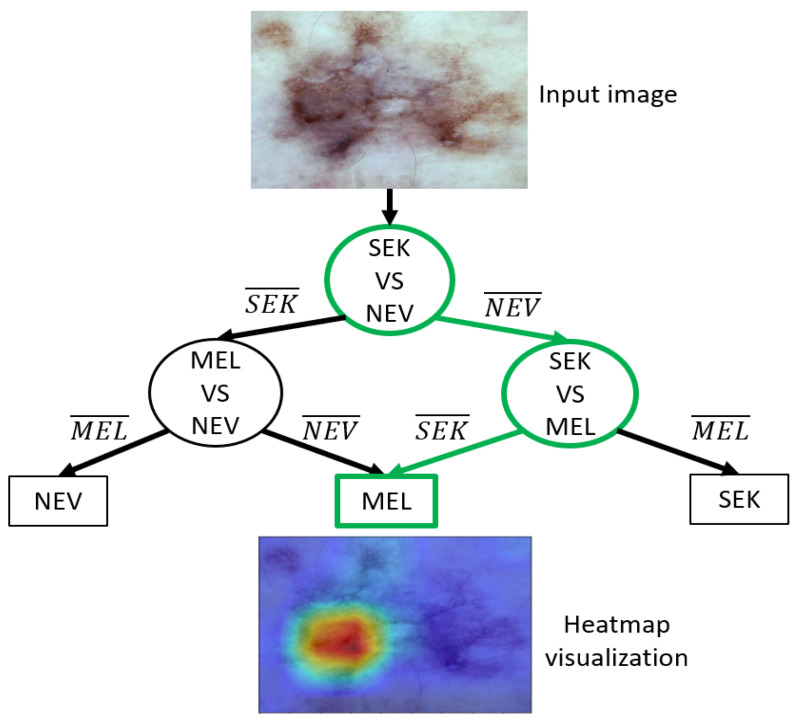
Output of our computer-aided diagnosis for melanoma detection. Heatmap generation was implemented with Grad-CAM [40]. MEL: melanoma, SEK: seborrheic keratosis, NEV: nevi.

**Table 1 sensors-21-03999-t001:** Overview of the related studies using ensemble methods with CNNs for skin disease recognition.

Authors	CNNs	Aggregation Method
[17]	GoogleNet, AlexNet, ResNet50, VGG16	Weighted majority voting
[18]	13 CNN models	Average probability
[19]	-	Average probability
[20]	AlexNet, VGG16, and resnet18	Average score
[21]	VGG-Net, ResNet50, InceptionV3, Xception, and DenseNet121	Average probability
[22]	-	Average probability
[23]	ResNet-50 and Inception V3	Average probability
[24]	Densenet, ResNeXt, PolyNet, and SENets	Average probability
[25]	10 CNN models	Geometric averaging

**Table 2 sensors-21-03999-t002:** Distribution of the dataset.

	Melanoma	Nevi	Seborrheic Keratosis
ISIC 2018	1113	6705	1099
Ratio	0.12	0.75	0.12
Original training set (80%)	779	4694	769
Test set (20%)	223	1341	220
Training set *	623	3755	615
Generated data from training set *	1377	1245	1385
Final training set with data generated *	2000	5000	2000
Ratio after data generation *	0.22	0.55	0.22
Validation set *	156	939	154

* Applied to each fold.

**Table 3 sensors-21-03999-t003:** Hyperparameters search space for finetuning.

Optimizer	Mini-Batch Size	Epoch	Ratio of Last Frozen Layers	Learning Rate
Adam	32	150	[4%, 65%]	[$10^{-2}$, $10^{-4}$ ]

**Table 4 sensors-21-03999-t004:** Binary CNNs’ performance on the training set with 3-fold cross-validation.

Task	CNN	BACC	S-MEL	S-NEV	S-SEK
MEL vs. NEV	VGG19_2	80.6 ± 0.3%	78.3 ± 3.0%	84.0 ± 3.5%	-
MEL vs. SEK	VGG19_2	85.8 ± 1.6%	85.6 ± 1.8%	-	84.5 ± 2.2%
NEV vs. SEK	VGG19_2	86.7 ± 1.2%	-	88 ± 0.8%	85.3 ± 2.3%
MEL vs. NEV	VGG16_2	80.2 ± 0.5%	75.0 ± 1.0%	84 ± 1.0%	-
MEL vs. SEK	VGG16_2	83.4 ± 1.9 %	87 ± 2.0%	-	79.5 ± 0.5%
NEV vs. SEK	VGG16_2	87.2 ± 0.4%	-	83.5 ± 1.5%	90.5 ± 1.5%
MEL vs. NEV	ResNet50_2	81.83 ± 1.63%	82.3 ± 0.4%	81.2 ± 2.6%	-
MEL vs. SEK	ResNet50_2	81.34 ± 0.51%	90 ± 1.0%	-	71.5 ± 1.9%
NEV vs. SEK	ResNet50_2	81.45 ± 0.4%	-	87 ± 1.3%	76 ± 1.5%

**Table 5 sensors-21-03999-t005:** Effects of varying the root node on the DDAG. The best performance achieved is highlighted in bold.

Root Node	CNN	BACC
MEL vs. NEV	VGG19_2	73.7 ± 1.2%
MEL vs. SEKL	VGG19_2	74.15 ± 0.94%
**NEV vs. SEK **	VGG19_2	**76.6 ± 0.39**
MEL vs. NEV	VGG16_2	72.55 ± 1.64%
MEL vs. SEK	VGG16_2	72.9 ± 1.69%
**NEV vs. SEK**	VGG16_2	**73.25 ± 1.54%**
**MEL vs. NEV**	ResNet50_2	**71.1 ± 0.6%**
MEL vs. SEK	ResNet50_2	70.1 ± 1.0%
NEV vs. SEK	ResNet50_2	70.35 ± 0.75%

**Table 6 sensors-21-03999-t006:** Comparison between the best model of DDAG and multiclass CNNs.

	BACC
DDAG with VGG19_2	76.6 ± 0.39%
VGG19_3	76.52 ± 1.5%
DDAG with VGG16_2	73.25 ± 1.54%
VGG16_3	72.52 ± 1.5%
DDAG with ResNet50_2	71.1 ± 0.6%
ResNet50_3	70.93 ± 1.08%

**Table 7 sensors-21-03999-t007:** Statistical comparison of DDAGs and multiclass CNNs in test set.

CNN	BACC		*p* < 0.05
	DDAG	3-class CNN	
VGG19	76.6%	74.6%	***
VGG16	73.9%	73.6%	***
Resnet50	70.4%	62.4%	***

***: *p* value < 0.0001.

**Table 8 sensors-21-03999-t008:** Comparison of our approach with classical aggregation methods and other related methods on the same task. The group rank of methods with statistically similar BACC is shown in “g.r” (1 is the best), while “s.o.g” shows which groups are statistically worse. The statistical test used there is Kruskal–Wallis’s test and Dunn’s multiple-comparison test.

Aggregated Method	CNN	BACC	Statistical Test	
			g.r	s.o.g
Our approach	VGG19_2	76.6%	1	{2–4}
AVG	ResNet50_3+ VGG16_3 + VGG19_3	75.2%	2	{3–4}
Max_conf	ResNet50_3+ VGG16_3 + VGG19_3	75.2%	2	{3–4}
Gmean	ResNet50_3+ VGG16_3 + VGG19_3	75.2%	2	{3–4}
Max-win	ResNet50_3+ VGG16_3 + VGG19_3	74.6%	3	{4}
[41]	VGG16	74.3%	-	-
Prod	ResNet50_3+ VGG16_3 + VGG19_3	73.9%	4	{}
[42]	DenseNet-161	70%	-	-
			p<0.05	

## Data Availability

The dataset is available online at https://challenge2018.isic-archive.com/task3/training/ (accessed on 8 June 2021).

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
