# Peer review of "Ensemble Method of Convolutional Neural Networks with Directed Acyclic Graph Using Dermoscopic Images: Melanoma Detection Application"

_sensors, 2021, doi:10.3390/s21123999_

Round 1

Reviewer 1 Report

The manuscript entitled “Ensemble method of convolutional neural networks with directed acyclic graph using dermoscopic images: application on melanoma detection.” reports about the development of an innovative ensemble scheme of convolutional neural networks and ensemble methods to improve the Computer Aided Diagnosis systems performances in classifying melanoma, nevi and seborrheic keratosis. The paper is very interesting and well written, however some concerns need to be addressed before publication:

MAJOR:

  • The Authors state that their method outperforms the performances of previous methods and other 3-levels classifiers. I suggest performing statistical comparisons between the accuracy and the AUC of the different classifiers considered in order to assess if the improvement in the performances is statistically significant.
  • The Authors state that they employ an “an artificial data generation” to balance the classes numerosity. Please specify the final class numerosity used to train and test the classifiers, after the data augmentation procedure.

MINORS:

  • Please, define the acronym only once in the manuscript. For instance, CAD and DAG are explained several times in the manuscript. Moreover, do DAG and DDAG indicate different approaches? If they are indicative of the same approach, please uniform with only one acronym.
  • Lines 111-117: In order to improve the readability of the manuscript, please rewrite the paragraph stating the authors of the cited manuscript and in the end the reference. For instance, line 111 will be paraphrased as follow: “ Harangi combined the output of the classification layer from four CNNs using a weighted majority voting strategy for a three-class classification task [12].”
  • Line 267, please correct “thapurpose”
  • Table 5, please correct “roof node”

Reviewer 2 Report

The manuscript present an ensemble approach of CNN to classify dermoscopic images. The structure is well organized, with clear content.

However, with classification of 3 classes, the highest accuracry only achive 76.6% is not high, it is difficult to apply to real would application. How can you improve more accuracy?

minor english spelling also need to check such as line 267.

Reviewer 3 Report

The manuscript submitted by Gouabou and co-authors describes a pipeline for the analysis of skin lesion images with image processing techniques. The manuscript is presented in a rather careless way, with numerous errors that will be mentioned below, with not very good English and some peculiarities, like some things written in French (Courbe ROC). The manuscript is difficult to read and this impairs to read into the possible contributions from the authors who claim to outperform other methodologies in the literature.

Structure of the manuscript

It would be better to follow the structure of Abstract, Introduction, Materials and Methods, Results and Discussion. At the moment, everything is a bit mixed up. Just one example, in section 4.1 L281 “Early diagnosis of melanoma among nevi and seborrheic keratosis constitutes a daily-life problematic for onco-dermatologists [1–3].” This is part of an introduction. The related work should go inside the introduction, experimental setup should be part of a section called Materials and Methods (in that order, first describe the materials, then the methods). In materials is important to give details and show more images of the data set being analysed.

Specific comments per section:

Introduction

The introduction is rather short and is not very clear in the way it is written, specific comments below, but the main point that should be made here is why is the current literature insufficient. The paragraph of lines 44-53 convinces us of the need of CAD, and then we jump to the approach followed in the paper without a proper justification of why this new approach is necessary. A bit more information for non-dermatologist is also necessary, are there only 3 types of skin cases, i.e. melanoma, nevi and keratosis? I do not think so, but I am not a dermatologist.

Specific comments

L21 “Skin cancers is one of the most common cancers among Caucasian population.” Please provide a reference for this and state what you mean exactly by “one of the most common” more than breast or colorectal?

L22 “Melanoma is rare among skin cancers, however its mortality is high by the possible evolution into a metastasis stage” Similarly, how rare is rare? And how mortal it is? Add references.

L23 “Moreover, melanoma is frequently misled with others pigmented lesions such as nevi and seborrheic keratosis” This is not clear, perhaps instead of misled, the authors mean “confused with”

L29 “Dermoscopic sensor is a non-invasive…” this is not good English, it should be “A Dermoscopic sensor is a non-invasive…” this error is frequent in the manuscript, please proofread carefully. Other cases of this “different lighting scheme” different lighting schemeS, “THE  or A  Dermoscopic sensor helps”

L42 “… unfortunately, dermatologists are lacking to run …” unclear, perhaps they mean there are too few dermatologists.

L45 “The evolution of the computing capacity and the availability of public dataset conduced convolutional neural networks to  unanimously became the benchmark for skin lesion classification” please re-write.

Related work,

This section introduces well the data, but since the different lighting schemes were previously mentioned, examples of how the lesions look like with those schemes would be expected in Fig 2. In terms of the literature, it seems that all references are 2019 or earlier, given the speed at which the field is moving, it would be good to scan 2020, scholar returns 18,000 documents for skin lesion classification (https://scholar.google.co.uk/scholar?as_ylo=2020&q=skin+lesion+classification&hl=en&as_sdt=0,5) and there are 3,000+ in 2021. For skin lesion classification ensemble the numbers are also in the thousands, surely some of these would be relevant to the current document

Methods

L182 “Training pairwized CNNs” pairwized is a word that does not exist in English.

L192 “We applied standard preprocessing for deep learning classification such as normalization, cropping and resizing image. We also performed color standardization to 194 ensure the robustness of our algorithms. In fact, images on dataset [21] are collected from 195 multiple sources and acquired under different setups.” The description of the pipeline should be precise and factual. If this methodology is to be reproducible the steps should be absolutely clear, that is, it is not valid to say “such as”. The normalisation, cropping and all steps should be perfectly described, how was the cropping decided? How did the re-sizing affects the image? Surely a small lesion is different from a large one in clinical terms. How was the scale of acquisition taken into account?

The references are all mixed up, just to name some of the errors

L196 “We used the gray world [22] algorithm to perform color standardization of the images. As advised in [23], we modified …”

  1. Galar, M.; Fernández, A.; Barrenechea, E.; Bustince, H.; Herrera, F. Deep learning ensembles for melanoma recognition in dermoscopy images. Pattern Recognition 2011, 44 pp. 1761–1776.[CrossRef]
  2. Codella, N.; Rotemberg, V.; Tschandl, P.; Celebi, M.E.; Dusza, S.; et al. Skin lesion analysis toward melanoma detection 2018: A challenge hosted by the international skin imaging collaboration (isic)arXiv preprint arXiv:1902.03368 2019.[arXiv]

22 Van De Weijer, J.; Gevers, T.; Gijsenij, A. Deep learning ensembles for melanoma recognition in dermoscopy images. IEEE Transactions on image processing 2007, 16 pp. 2207–2214.[CrossRef]

  1. Gijsenij, A.; Gevers, T.; Van DeWeijer, J. Computational color constancy: Survey and experiments. IEEE Transactions on Image Processing 2011, 20 pp. 2475–2489.[CrossRef]

20 and 22 have the same title but different journals and authors, and they are actually both wrong as the paper with title “Deep learning ensembles for melanoma recognition in dermoscopy images” is authored by Codella and published in 2017, not 2007 (https://ieeexplore.ieee.org/abstract/document/8030303). This is really poor. I am not going to check the rest but from here I do not trust any of the references and probably the actual results of this paper.

Results and Discussion

L354-359 “Table 6 shows the evaluation of our main hypothesis … We can  thus conclude that decomposing a multi-class problem into binary problem reduce the  complexity of the initial problem and increase the overall performance.” The results presented in table 6 do not show a massive increase in performance, 76.6+-0.39  v  76.52+-1.5  are practically the same, especially given the confidence intervals and the same applies to the other results.  

Round 2

Reviewer 3 Report

The authors have addressed all the issues that I have previously raised. I would just like to highlight that the code should be released through the git pages, with a sufficiently clear Readme so that the research becomes reproducible. This will benefit all the community, including the authors as if their code is useful and clear, it will be used and then the manuscript will increase its citations.

As a corolary, if all those errors arose from submitting an early version of the manuscript, it should be a lesson for the authors not to submit manuscripts without thoroughly checking them, other journals may not give a second chance when this level of errors was detected.
